# Smoothing Nonlinear Variational Objectives with Sequential Monte Carlo

**Antonio Khalil Moretti** [*]
Department of Computer Science
Columbia University
amoretti@cs.columbia.edu

**Zizhao Wang** [*]
Department of Computer Science
Columbia University
zw2504@columbia.edu

**Luhuan Wu**
Data Science Institute
Columbia University
lw2827@columbia.edu

**Itsik Pe'er**
Department of Computer Science
Columbia University
itsik@cs.columbia.edu

## Abstract

The task of recovering nonlinear dynamics and latent structure from a population recording is a challenging problem in statistical neuroscience motivating the development of novel techniques in time series analysis. Recent work has focused on connections between Variational Inference and Sequential Monte Carlo for performing inference and parameter estimation on sequential data. Inspired by this work, we present a framework to develop Smoothed Variational Objectives (SVOs) that condition proposal distributions on the full time-ordered sequence of observations. SVO maintains both expressiveness and tractability by sharing parameters of the transition function between the proposal and target. We apply the method to several dimensionality reduction/expansion tasks and examine the dynamics learned with a quantitative metric. SVO performs favorably against the state of the art.

## 1 Introduction

Conductance based models of excitable cells are widely used in neuroscience to describe the spiking activity of individual neurons. It is thought that neural computation is explained by dynamics and that these dynamics often exist in a lower or higher dimensionality than that of the recorded neural populations (Paninski & Cunningham, 2017). To extract the information encoded in firing activity, it is paramount to develop models that allow for tractable analysis of neural data. For example, experimentalists who have access to a single-dimensional observation such as a voltage recording are faced with the task of recovering the multidimensional nonlinear latent dynamics and trajectories through a higher dimensional space that describe the system of interest.

There is a large body of work for inferring latent trajectories for data governed by nonlinear dynamics or observations (Archer et al., 2015; Krishnan et al., 2015; Pandarinath et al., 2017; Diaz et al., 2019). Variational Inference (VI) and Markov Chain Monte Carlo (MCMC) are two popular approaches for performing inference in latent variable models. Recently connections have been established between both methods that allow for performing inference by defining a flexible variational family of filtered distributions using Sequential Monte Carlo (SMC) (Le et al., 2018; Maddison et al., 2017; Naesseth et al., 2018a). Unlike filtering, smoothing refers to the ability to condition states and parameters on the full time-ordered sequence of observations. In this framework, we sketch a method to construct variational objectives using recursive backwards sampling algorithms (referred to as particle smoothers) in addition to parameterizing the encoding function on the full time ordered sequence of observations.

Recent literature argues that tighter variational bounds may hurt the proposal learning (Rainforth et al., 2018). More specifically, in the context of the Importance Weighted AutoEncoder (IWAE),

---

[*]Equal contribution

the marginal likelihood estimate becomes exact as the number of particles $K \to \infty$. This can lead to a degenerated gradient estimate for the proposal with the signal-to-noise ratio shrinking at the rate $\mathcal{O}(\sqrt{1/K})$ (Rainforth et al., 2018). (Le et al., 2018) built on this idea to use different objectives for recognition and generative networks. Here we argue that the theoretical result in the IWAE setting cannot be directly extended to SMC due to the resampling process. We present theoretical and empirical evidence to show that the degrading signal-to-noise ratio is addressed by the choice of biased gradient estimators.

The contributions of this paper are highlighted below. First, we describe a factorized proposal distribution that (i) disentangles the transition terms from the terms that represent the encoding of the data and (ii) shares variational parameters of the transition function with the target distribution. We present results illustrating the positive effect of the number of particles on the signal-to-noise ratio for the transition, encoder and decoder parameters within our factorization. An important way to ascertain the quality of the learned dynamics is to use the transition terms in the generative model to propagate the system forwards without input data and then to make observation predictions. We show that our smoothed proposal generates an improved estimate of the latent states as measured by the ability of the target to more accurately predict observations using the dynamics learned.

The remainder of this paper is organized as follows. Section 2 summarizes related work on VI and SMC. Section 3 gives preliminaries and defines the proposal while analyzing the signal to noise ratio of the gradient. We then present results on both filtered and smoothed proposal densities with two benchmark datasets and single cell electrophysiology data from the Allen Institute in Section 4. Section 5 concludes.

## 2 RELATED WORK

There is a large body of work on fitting State Space Models (SSMs) of increasing complexity using VI. These methods can be separated into two classes. Pure VI methods optimize a lower bound to the log likelihood parameterized by defining a variational family. Hybrid methods use SMC to construct an estimate for the log marginal likelihood and optimize a surrogate lower bound. We review the details of both techniques in the preliminaries section and outline the contributions of related methods below.

Gaussian Process Factor Analysis (Yu et al., 2009) (GPFA) is a statistical model that assumes linear time invariant transition and emission functions. GfLDS (Gao et al., 2016; Archer et al., 2015) is a VI scheme that extends the above by positing a linear dynamical system with nonlinear emission densities. VIND (Diaz et al., 2019) is a VI scheme that permits both transition and emission densities to be represented as arbitrary nonlinearities. VIND achieves reuse of the exact generative evolution structure for inference as suggested by the true posterior. Tractability of the variational approximation is maintained by forming a Laplace approximation to an intractable non Gaussian parent approximate distribution. The mean and the covariance of the Laplace approximation are obtained via the Fixed Point Iteration (FPI) and the intractable terms are reused within the approximation. We define smoothing as a method to infer latent state $\mathbf{z}_t$ using observations from the complete trial $\mathbf{x}_{1:T}$, including points to the future of $t < t' \in \{t+1 : T\}$. GPFA, GfLDS and VIND are all smoothers in that they use the full sequence of observations to infer the current latent state.

An approach to increasing model capacity is the development of tighter bounds than the ELBO. Filtering Variational Objectives (Maddison et al., 2017) (FIVO) and Variational Sequential Monte Carlo (Naesseth et al., 2018a) are related methods that use SMC to construct a filtered objective function for VI. Auto-Encoding Sequential Monte Carlo (Le et al., 2018) (AESMC) is a similar method to learn both proposal distributions for SMC that act as variational approximations and generative models using deep neural networks. Unlike GfLDS and VIND, these methods do not require inverting a block-tridiagonal matrix which mixes components of state space through the inverse covariance. Information from the complete time ordered sequence $1 : T$ is not applied directly to the transition function to the future of the current time point $t < T$ to infer the latent state. TVSMC (Lawson et al.) is a family of variational objectives that augment the target distribution with a lookahead function to approximate smoothing SMC. Temporal difference learning is used to estimate the lookahead function, however this method did not produce results comparable with filtering on complex nonlinear models.

## 3 METHODS

### 3.1 PRELIMINARIES

#### 3.1.1 VI AND IMPORTANCE WEIGHTED AUTO-ENCODERS

Inference in latent variable models requires marginalizing a generative distribution with respect to hidden variables $\mathbf{Z}$.

$$\log p_\theta(\mathbf{X}) = \int \log p_\theta(\mathbf{X}, \mathbf{Z}) d\mathbf{Z} \tag{1}$$

VI describes family of techniques for approximating a solution to the above. The idea is to define a tractable distribution $q_\phi(\mathbf{Z}|\mathbf{X})$ and to optimize a lower bound to the log likelihood:

$$\log p_\theta(\mathbf{X}) \geq \mathcal{L}_{\text{ELBO}}(\theta, \phi, \mathbf{X}, \mathbf{Z}) = \mathbb{E}_q \left[ \log \frac{p_\theta(\mathbf{X}, \mathbf{Z})}{q_\phi(\mathbf{Z}|\mathbf{X})} \right] \tag{2}$$

Tractability and expressiveness of the variational approximation $q_\phi(\mathbf{Z}|\mathbf{X})$ are contrasting goals. The distribution $q_\phi(\mathbf{Z}|\mathbf{X})$ must be chosen carefully; a poor choice of the variational family defining $q_\phi(\mathbf{Z}|\mathbf{X})$ will affect the generative model by pulling the parameters $\theta$ towards $\phi$.

$$\theta^*, \phi^* = \underset{\theta \in \Theta, \phi \in \Phi}{\arg\min} \, D_{\text{KL}}(q_\phi || p_\theta) = \underset{\theta \in \Theta, \phi \in \Phi}{\arg\max} \, \mathcal{L}_{\text{ELBO}} \tag{3}$$

Auto Encoding Variational Bayes (Kingma & Welling, 2013) (AEVB) is method to simultaneously encode and decode the parameters $\{\phi, \theta\}$ for both $q_\phi(\mathbf{Z}|\mathbf{X})$ and $p_\theta(\mathbf{X}, \mathbf{Z})$. When the densities are not in the natural exponential family, it is possible to draw samples from $q_\phi(\mathbf{Z}|\mathbf{X})$ to evaluate $\mathbb{E}_{q_\phi}[\log p_\theta(\mathbf{X}, \mathbf{Z}) - \log q_\phi(\mathbf{Z}|\mathbf{X})]$ numerically. We can approximate both $\mathcal{L}_{\text{ELBO}}$ terms reparameterizing in order to compute the gradients (Kingma & Welling, 2013; Rezende et al., 2014) as follows:

$$\mathcal{L}_{\text{AEVB}}(\theta, \phi, \mathbf{X}^{(i)}) \approx \frac{1}{L} \sum_{l=1}^{L} \log \frac{p_\theta(\mathbf{X}^{(i)}, \mathbf{Z}^{(i,l)})}{q_\phi(\mathbf{Z}^{(i,l)}|\mathbf{X}^{(i)})} \tag{4}$$

$$\text{where} \quad \mathbf{Z}^{(i,l)} = \mu_\phi + \Sigma_\phi^{1/2} \epsilon^{(i,l)} \quad \text{and} \quad \epsilon^{(i,l)} \sim \mathcal{N}(0, 1) \tag{5}$$

Building upon this, the Importance Weighted Auto Encoder (Burda et al., 2015) (IWAE) constructs tighter bounds than the $\mathcal{L}_{\text{AEVB}}$ through mode averaging as opposed to mode matching. The idea is to estimate $\mathcal{L}_{\text{AEVB}}$ by drawing $K$ samples from a proposal distribution and to average probability ratios:

$$\mathcal{L}_{\text{IWAE}}^{K}(\theta, \phi, \mathbf{X}, \mathbf{Z}) \approx \log \frac{1}{K} \sum_{k=1}^{K} \frac{p_\theta(\mathbf{X}, \mathbf{Z}^{(k)})}{q_\phi(\mathbf{Z}^{(k)}|\mathbf{X})} \qquad \text{where} \quad \mathbf{Z}^{(k)} \sim q_\phi(\mathbf{Z}|\mathbf{X}) \tag{6}$$

It can be shown that $\log p_\theta(\mathbf{X}) \geq \mathcal{L}^{k+1} \geq \mathcal{L}^k \geq \mathcal{L}^1$. The $\mathcal{L}_{\text{IWAE}}$ reduces to the $\mathcal{L}_{\text{AEVB}}$ when $K = 1$ and approaches the true log probability of the data as $K \to \infty$.

#### 3.1.2 STATE SPACE MODELS

State space models (SSMs) describe a time series of observations $\mathbf{X} \equiv \{\mathbf{x}_1, \ldots \mathbf{x}_T\}$, $\mathbf{x}_t \in \mathbb{R}^{d_\mathbf{x}}$ dependent upon a time series of latent variables $\mathbf{Z} \equiv \{\mathbf{z}_1, \ldots \mathbf{z}_T\}$, $\mathbf{z}_T \in \mathbb{R}^{d_\mathbf{z}}$ that evolve according to stochastic dynamics. We are interested in dynamical systems which exhibit the following dependency:

$$p_\theta(\mathbf{X}, \mathbf{Z}) = c_\theta \cdot F_\theta(\mathbf{Z}) \prod_{t=0}^{T} g_\theta(\mathbf{x}_t|\mathbf{z}_t), \tag{7}$$

where $g_\theta$ is an observation model whose parameters are functions of the latent state $\mathbf{z}_t$. The normalization constant is denoted $c_\theta$ and $F_\theta$ denotes the latent evolution described with Markov Chain:

$$F_\theta(\mathbf{Z}) = f_0(\mathbf{z}_0) \prod_{t=1}^{T} f_\theta(\mathbf{z}_t|\mathbf{z}_{t-1}), \tag{8}$$

$$f_0 = \mathcal{N}(\psi_0, \mathbf{Q}_0), \tag{9}$$

$$\mathbf{z}_t|\mathbf{z}_{t-1} \sim \mathcal{N}(\psi_\theta(\mathbf{z}_{t-1}), \mathbf{Q}), \tag{10}$$

Marginalization with respect to $\mathbf{z}$ becomes intractable when $F_\theta(\mathbf{Z})$ is defined by a nonlinear function or $g(\mathbf{x}_t|\mathbf{z}_t)$ is non-Gaussian.

### 3.1.3 SEQUENTIAL MONTE CARLO

Sequential Monte Carlo (SMC) methods factorize an intractable distribution $p_\theta(\mathbf{X}, \mathbf{Z})$ (referred to as the target distribution) into a distribution of increasing probability spaces. Sequential Importance Sampling (SIS) methods perform importance sampling sequentially by defining importance weights $w_t^{(k)}$ at time $t$ for sample $k$ as follows where $K$ samples are drawn from the proposal distribution $q$:

$$\mathbf{z}_t^{(k)} \sim q_\phi(\mathbf{z}_t^{(k)}|\mathbf{z}_{t-1}^{(k)}, \mathbf{x}_t) \qquad w_t^{(k)} := \frac{f_\theta(\mathbf{z}_t^{(k)}|\mathbf{z}_{t-1}^{(k)})g_\theta(\mathbf{x}_t|\mathbf{z}_t^{(k)})}{q_\phi(\mathbf{z}_t^{(k)}|\mathbf{z}_{t-1}^{(k)}, \mathbf{x}_t)} \tag{11}$$

Various resampling schemes exist so that the samples which are referred to as particles are focused on promising regions of state space. Sequential Importance Resampling and SMC achieve this goal by resampling particles according to their importance sampling weights:

$$\mathbf{a}_{t-1}^{(k)} \sim \text{CATEGORICAL}(\cdot|\mathbf{w}_{t-1}^{(1:K)}) \qquad w_t^{(k)} := \frac{f_\theta(\mathbf{z}_t^{(k)}|\mathbf{z}_{t-1}^{\mathbf{a}_{t-1}^{(k)}})g_\theta(\mathbf{x}_t|\mathbf{z}_t^{\mathbf{a}_{t-1}^{(k)}})}{q_\phi(\mathbf{z}_t^{(k)}|\mathbf{z}_{t-1}^{\mathbf{a}_{t-1}^{(k)}}, \mathbf{x}_t)} \tag{12}$$

At the last time step, the posterior can be evaluated by averaging over sample trajectories to approximate the functional integral using the empirical measure below:

$$\sum_{k=1}^{K} \bar{w}_T^{(k)} \delta_{\mathbf{z}_{1:T}^{(k)}}(\mathbf{z}_{1:T}) \qquad \text{where} \qquad \bar{w}_T^{(k)} = w_T^{(k)} / \sum_{j=1}^{K} w_T^{(j)} \tag{13}$$

The weights can now be used to construct an unbiased estimate for the marginal likelihood:

$$\hat{\mathcal{Z}}_{\text{SMC}} := \prod_{t=1}^{T} \left[ \frac{1}{K} \sum_{k=1}^{K} w_t^{(k)} \right] \tag{14}$$

### 3.1.4 AUTO ENCODING SEQUENTIAL MONTE CARLO

An important insight of (Maddison et al., 2017; Le et al., 2018) is that the SMC algorithm is deterministic conditioning on $(\mathbf{Z}_{1:T}^{(1:K)}, \mathbf{A}_{1:T-1}^{(1:K)})$. As a result, the importance sampling density can be reparameterized to act as a variational distribution that can be encoded:

$$Q_{\text{SMC}}(\mathbf{Z}_{1:T}^{1:K}, \mathbf{A}_{1:T-1}^{1:K}) := \left( \prod_{k=1}^{K} q_{1,\phi}(\mathbf{z}_1^{(k)}) \right) \prod_{t=2}^{T} \prod_{k=1}^{K} q_{t,\phi}(\mathbf{z}_t^{(k)}|\mathbf{z}_{1:t-1}^{\mathbf{a}_{t-1}^{(k)}}) \cdot \text{CATEGORICAL}(\mathbf{a}_{t-1}^{(k)}|\mathbf{w}_{t-1}^{1:K}) \tag{15}$$

This gives a way of constructing a filtered cost function for simultaneous model inference and learning. The cost is constructed by running SMC to obtain an estimate of the marginal log likelihood. As with the IWAE, the estimate $\mathcal{L}_{\text{SMC}}$ defined below converges to the true log likelihood $\mathcal{L}_{\text{SMC}} \to \log p(\mathbf{x}_{1:T})$ as $K \to \infty$.

$$\mathcal{L}_{\text{SMC}}(\theta, \phi, \mathbf{Z}_{1:T}) := \int Q_{\text{SMC}}(\mathbf{Z}_{1:T}^{1:K}, \mathbf{A}_{1:T-1}^{1:K}) \log \hat{\mathcal{Z}}_{\text{SMC}}(\mathbf{Z}_{1:T}^{1:K}, \mathbf{A}_{1:T-1}^{1:K}) d\mathbf{Z}_{1:T}^{1:N} d\mathbf{A}_{1:T}^{1:K} \tag{16}$$

## 3.2 SMOOTHING VARIATIONAL OBJECTIVES

We build upon AESMC and FIVO to design a variational objective based on the marginal likelihood estimate constructed using SMC. One way to define a Smoothed Variational Objective (SVO) is to explicitly modify the target density by conditioning latent states on future observations. Our approach is to smooth the proposal distribution by conditioning on full observations thereby implicitly modifying the target through its shared factorization with the proposal.

### 3.2.1 PARAMETERIZING THE PROPOSAL DISTRIBUTION

We begin by considering a proposal distribution of the form:

$$q_{\phi,\varphi}(\mathbf{z}_{1:T}^{(k)}|\mathbf{x}_{1:T}) \propto \underbrace{q_\varphi(\mathbf{z}_1^{(k)})}_{initial\ state} \prod_{t=1}^{T} \underbrace{q_\phi(\mathbf{z}_t^{(k)}|\mathbf{x}_t)}_{encoding} \prod_{t=2}^{T} \underbrace{\text{CATEGORICAL}(\mathbf{a}_{t-1}^{(k)}|\mathbf{w}_{t-1}^{1:K})}_{resampling} \underbrace{q_\varphi(\mathbf{z}_t^{(k)}|\mathbf{z}_{t-1}^{a_{t-1}^{(k)}})}_{evolution} \tag{17}$$

where the proposal density factorizes into separate functions for evolution of the latent dynamics and encodings of the data.

$$q_\varphi(\mathbf{z}_t|\mathbf{z}_{t-1}) = \mathcal{N}(\psi(\mathbf{z}_{t-1}), \Sigma), \tag{18}$$

$$q_\phi(\mathbf{z}_t|\mathbf{x}_t) = \mathcal{N}(\gamma(\mathbf{x}_t), \Lambda). \tag{19}$$

We take $\psi : \mathbb{R}^{d_\mathbf{z}} \to \mathbb{R}^{d_\mathbf{z}}$ and $\gamma : \mathbb{R}^{d_\mathbf{x}} \to \mathbb{R}^{d_\mathbf{z}}$ as nonlinear time invariant functions represented with deep neural networks. The covariances $\Sigma$ and $\Lambda$ can be invariant trainable parameters or nonlinear functions of the latent space. Unlike $Q_{\text{SMC}}$ in Eq. (15), Eq. (17) models dynamics that depend on latent states instead of time. This proposal choice is also advantageous because the transition term of the recognition model $q_\varphi(\mathbf{z}_t^{(k)}|\mathbf{z}_{t-1}^{(k)})$ can be chosen to share the network parameters $\varphi$ defining $\{\psi, \Sigma\}$ with the target transition term $f_\varphi(\mathbf{z}_t|\mathbf{z}_{t-1})$ of the generative model.

$$p_{\theta,\varphi}(\mathbf{z}_{1:T}, \mathbf{x}_{1:T}) \propto \underbrace{f_\varphi(\mathbf{z}_1)}_{initial\ state} \prod_{t=1}^{T} \underbrace{g_\theta(\mathbf{x}_t|\mathbf{z}_t)}_{decoding} \prod_{t=2}^{T} \underbrace{f_\varphi(\mathbf{z}_t|\mathbf{z}_{t-1})}_{evolution} \tag{20}$$

This is analogous to the bootstrap filter (Gordon et al., 1993), however the addition of a new term permits disentangling the transition function from an encoding of the data. As a result the evolution term of the variational posterior is exact retaining both tractability and expressiveness. The generative evolution law is thus specified as follows:

$$f_\varphi(\mathbf{z}_t|\mathbf{z}_{t-1}) = \mathcal{N}(\psi(\mathbf{z}_{t-1}), \Sigma), \tag{21}$$

The decoding term is defined using a deterministic nonlinear rate function $\upsilon : \mathbb{R}^{d_\mathbf{z}} \to \mathbb{R}^{d_\mathbf{x}}$ represented with a deep network and a noise model that need not be conjugate. Without loss of generality consider a Gaussian emission density:

$$g_\theta(\mathbf{z}_t|\mathbf{x}_t) = \mathcal{N}(\upsilon(\mathbf{x}_t), \Gamma), \tag{22}$$

The incremental weights are expressed using the following factorization:

$$w_t^{(k)} \propto \frac{f_\varphi(\mathbf{z}_t^{(k)}|\mathbf{z}_{t-1}^{\mathbf{a}_{t-1}^{(k)}}) g_\theta(\mathbf{x}_t|\mathbf{z}_t^{\mathbf{a}_{t-1}^{(k)}})}{q_\varphi(\mathbf{z}_t^{(k)}|\mathbf{z}_{t-1}^{\mathbf{a}_{t-1}^{(k)}}) q_\phi(\mathbf{z}_t^{(k)}|\mathbf{x}_t)} \tag{23}$$

### 3.2.2 FORWARD FILTERING BACKWARD SMOOTHING

We consider two solutions for sampling smoothed trajectories, one is a bidirectional encoding architecture for smoothing and another is the Forward Filtering Backwards Smoothing (FFBS) formula (Kitagawa, 1996). To smooth the latent states and ensure that $q(\mathbf{z}_t)$ depends on both $\mathbf{z}_{t-1}$ and $\mathbf{z}_{t+1}$, we consider the following recursion:

$$p(\mathbf{z}_t|\mathbf{x}_{1:T}) = \underbrace{p(\mathbf{z}_t|\mathbf{x}_{1:t})}_{filtered} \int \frac{\overbrace{p(\mathbf{z}_{t+1}|\mathbf{x}_{1:T})}^{smoothed} \overbrace{p(\mathbf{z}_{t+1}|\mathbf{z}_t)}^{evolution}}{\underbrace{\int p(\mathbf{z}_{t+1}|\mathbf{z}_t) p(\mathbf{z}_t|\mathbf{x}_{1:t}) d\mathbf{z}_t}_{state\ prediction}} d\mathbf{z}_{t+1} \tag{24}$$

FFBS first iterates forward to compute the filtered distribution at each time step, reweighting particles with the following backward recursion:

$$w_{t|T}^{(k)} = \bar{w}_t^{(k)} \left[ \sum_{i=1}^{K} w_{t+1|T}^{(i)} \frac{f_\theta(\mathbf{z}_{t+1}^{(i)}|\mathbf{z}_t^{(k)})}{\sum_{j=1}^{K} \bar{w}_t^{(j)} f_{\theta,\varphi}(\mathbf{z}_{t+1}^{(i)}|\mathbf{z}_t^{(j)})} \right] \tag{25}$$

with $w_{T|T}^{(k)} = \bar{w}_T^{(k)}$. This allows us to sample from a smoothed posterior. When applying FFBS, we find two drawbacks. In contrast to the $\mathcal{O}(TK)$ complexity of particle filter, the complexity of FFBS is $\mathcal{O}(TK^2)$, scaling quadratically with the number of particles through the through the $(i, k)$ pairwise interactions. Another limitation includes the fact that the backwards recursion to update weights does not change the support of the sampled particles which may be disjoint from the smoothed posterior.

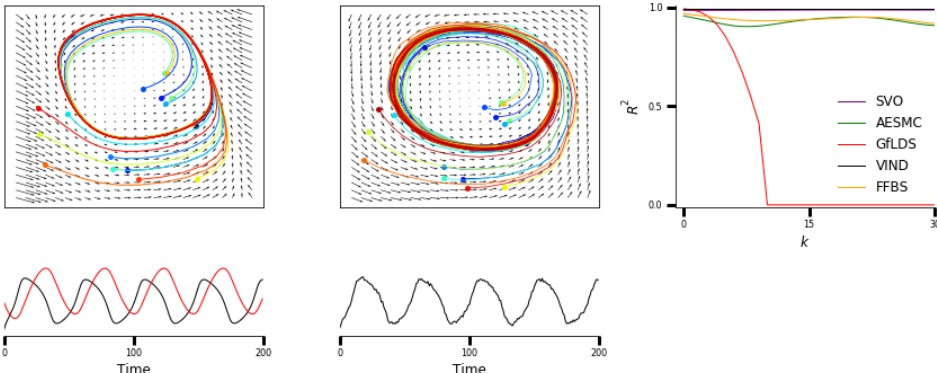

Figure 1: Summary of the Fitzhugh-Nagumo results: (top left) ground truth dynamics and trajectories for the original system; (bottom left) latent variables $\mathbf{z}_t = (V_t, W_t)$ from a single trial in the plane; (top center) latent dynamics and trajectories inferred by SVO; (bottom center) noisy 1D observation $\mathbf{x}_t = \mathcal{N}(V_t, \sigma^2)$ from a single trial; (right) $R_k^2$ for various models on the dimensionality expansion task.

### 3.2.3 ENCODING ARCHITECTURES FOR SMOOTHED VARIATIONAL OBJECTIVES

We augment the encoding term of the proposal in Eq. (17) using the full time ordered sequence of observations:

$$Q_{\text{SVO}}(\mathbf{z}_{1:T}^{(k)}|\mathbf{x}_{1:T}) \propto q_\varphi(\mathbf{z}_1^{(k)}|\mathbf{x}_{1:T}) \prod_{t=1}^{T} q_\phi(\mathbf{z}_t^{(k)}|\mathbf{x}_{1:T}) \prod_{t=2}^{T} \text{CATEGORICAL}(\mathbf{a}_{t-1}^{(k)}|\mathbf{w}_{t-1}^{1:K}) q_\varphi(\mathbf{z}_t^{(k)}|\mathbf{z}_{t-1}^{(k)}{}^{\mathbf{a}_{t-1}^{(k)}})$$

(26)

For the case where $q_{t,\phi}(\mathbf{z}_1|\mathbf{x}_1)$ and $q_{t,\phi}(\mathbf{z}_t|\mathbf{x}_{1:t})$ are factorized Gaussians with conditionals parameterized by time varying networks, simply expanding the network inputs from $\mathbf{x}_t$ to $\mathbf{x}_{1:T}$ would be expensive. As a result, we extract features from the full time series $\mathbf{x}_{1:T}$ and redefine the input to our encoder function. To achieve this, an RNN is run both forward and backward,

$$e_{f,t} = \text{RNN}_f(e_{f,t-1}, \mathbf{x}_t) \tag{27}$$
$$e_{b,t} = \text{RNN}_b(e_{b,t+1}, \mathbf{x}_t) \tag{28}$$

where $e_{f,t}$ and $e_{b,t}$ are states containing information of $\mathbf{x}_{1:t}$ and $\mathbf{x}_{t:T}$ respectively. Then the concatenation of $[e_{f,T}, e_{b,1}]$ is used to parameterize the initial state distribution $q_\phi(\mathbf{z}_1|\mathbf{x}_{1:T})$, and $[e_{f,t}, e_{b,t}]$ is fed into the term of the proposal defining the encoder $q_\phi(\mathbf{z}_t|\mathbf{x}_{1:T})$. The nonlinear time invariant evolution function is then applied to transition between states.

$$\mathbf{z}_1 = \gamma([e_{f,T}, e_{b,1}]) \tag{29}$$
$$\mathbf{z}_t = \psi(\mathbf{z}_{t-1}) \tag{30}$$

The cost of the feature extraction with the bidirectional RNN is $\mathcal{O}(T)$, independent of the number of particles unlike the FFBS. This is similar to an approach in FIVO with the exception of a shared parameterization between proposal and target transition. The objective and marginal likelihood estimate are defined below.

$$\mathcal{L}_{\text{SVO}} := \mathbb{E}_{Q_{\text{SVO}}}\left[\log \hat{\mathcal{Z}}_{\text{SVO}}\right] \quad \text{where} \quad \hat{\mathcal{Z}}_{\text{SVO}} := \prod_{t=1}^{T} \frac{1}{K} \sum_{k=1}^{K} \frac{f_\varphi(\mathbf{z}_t^{(k)}|\mathbf{z}_{t-1}^{(k)}{}^{\mathbf{a}_{t-1}^{(k)}}) g_\theta(\mathbf{x}_t|\mathbf{z}_t^{(k)}{}^{\mathbf{a}_{t-1}^{(k)}})}{q_\varphi(\mathbf{z}_t^{(k)}|\mathbf{z}_{t-1}^{(k)}{}^{\mathbf{a}_{t-1}^{(k)}}) q_\phi(\mathbf{z}_t^{(k)}|\mathbf{x}_{1:T})} \tag{31}$$

### 3.2.4 NONLINEAR EVOLUTION OF THE COVARIANCE MATRIX.

For many nonlinear dynamical systems, the evolution of the latent state may depend on a noise term which itself may evolve stochastically. In order to model a covariance matrix that expresses nonlinear $\mathbf{z}$-dependence on the latent space, we face a challenge. It is important to ensure that the latent paths are smooth. This is equivalent to stating that the difference between the covariance matrix and a constant matrix $\mathbb{C}$ is small.

$$\max |\mathbf{Q}(\mathbf{z}_t) - \mathbb{C}| \lesssim 0.1 \tag{32}$$

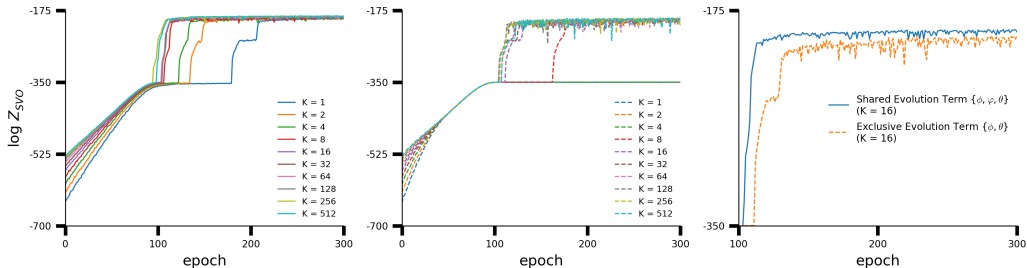

Figure 2: ELBO convergence across epochs for SVO using exclusive parameters $\theta, \phi$ and shared parameters $\theta, \varphi, \phi$; (left) $\log \mathcal{Z}_{SVO}$ across epochs as $K$ increases using shared evolution network; (center) $\log \mathcal{Z}_{SVO}$ across epochs as $K$ increases using independent evolution networks; (right) $\log \mathcal{Z}_{SVO}$ convergence for shared vs independent evolution networks with $K = 16$ highlighting faster convergence to a higher ELBO.

An expressive model should permit the covariance to undergo nonlinear evolution. One solution is to parameterize the covariance with a constant plus a scalar times a symmetric matrix whose components are a nonlinear function of the latent state.

$$\mathbf{Q}(\mathbf{z}_t) = \mathbb{C} + \alpha \cdot \mathbf{\Sigma}(\mathbf{z}_t) \tag{33}$$

In the experiments we take $\alpha = 1e - 1$, $\mathbb{C} = \mathbb{I} \cdot \sigma^2$ where $\sigma^2$ is a trainable variable and the components of $\mathbf{\Sigma}(\mathbf{z}_t)$ as the output of a deep network. This permits dynamic proposal distributions whose entropy evolves as the Markov chain transitions between states. At different positions in latent space, the system can suppress or enhance its sensitivity to noise. We refer to this as a Locally Linear Covariance Matrix (LLCM).

### 3.3 Gradient estimators and Signal-to-noise Ratio

Because $\mathcal{L}_{\text{SVO}}$ converges to the true log marginal likelihood, one would naturally think about increasing the number of particles $K$ to get a better surrogate objective. However, (Rainforth et al., 2018) points out the detrimental effect of large $K$ on learning the inference network by reducing the signal-to-noise-ratio of the gradient estimator. Formally, for a gradient estimator $\Delta_K$ constructed by $K$ particles, the signal-to-noise ratio (SNR) is defined as:

$$\text{SNR}_K = \left| \frac{\mathbb{E}[\Delta_K]}{\sqrt{\text{Var}[\Delta_K]}} \right|. \tag{34}$$

Intuitively, a vanishing SNR implies that the gradient estimator reduces to pure noise, hence providing no information to learning. In the IWAE setting, the SNR of the inference network decreases with rate $\mathcal{O}(1/\sqrt{K})$ (Rainforth et al., 2018); (Le et al., 2018) extends the result to the SMC setting without providing the theoretical evidence. Here, we argue that (Le et al., 2018) neglects that they are using a biased gradient estimator, which in fact addresses the issue of degrading SNR.

We consider the following three stochastic gradient estimators. A full description would be given in the Appendix.

1. The unbiased estimator, denoted by $\nabla \mathcal{L}_K + $ CATEGORICAL, which takes into account the gradient in the resampling procedure.

2. The biased estimator without resampling gradient $\nabla \mathcal{L}_K$

3. The relaxed gradient estimator, $\nabla \mathcal{L}_K + $ CONCRETE$(\lambda)$, which replaces the CATEGORICAL distribution in the resampling step with CONCRETE distribution. Note that as $\lambda \to 0$, the CONCRETE distribution approaches the CATEGORICAL. (Jang et al., 2016), (Maddison et al., 2016).

For the SNR of $\nabla \mathcal{L}_K$, we have the following proposition:

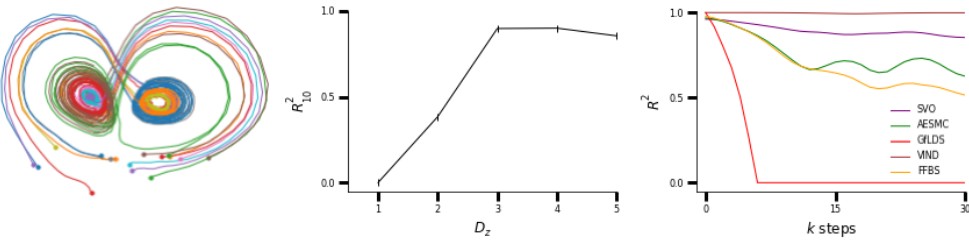

Figure 3: Summary of the Lorenz results: (left) inferred latent paths from noisy 10D observations; (center) the $R^2_{k=10}$ plot for ten step forward interpolation as latent dimension $d_\mathbf{z}$ increases; (right) the $R^2_k$ for $d_\mathbf{z} = 3$ across various models on the dimensionality reduction task.

**Proposition 1.** *Assume that the first four moments of $w_t^{(1)}$ and $\nabla w_t^{(1)}$ are all finite and their variances are non-zero for $t \in 1 : T$, then the signal-to-noise ratio converges at the following rate:*

$$
SNR_K(\theta, \varphi, \phi)
$$

$$
= \left| \frac{\nabla \log Z + \sum_{t=2}^{T} \sum_{t' \geq t+1}^{T} \mathbb{E}\left[ \nabla \frac{w_{t-1}^1}{Z_{t-1}} \cdot \frac{(w_{t'}^1 - Z_{t'})^2}{2Z_{t'}^2} \middle| (a_{t-1}^1 = 1) \right] + \mathcal{O}(1/K)}{\sqrt{1/K \left\{ \sum_{t=1}^{T} \mathbb{E}\left[ (\nabla \frac{w_t^1}{Z_t})^2 \right] + \sum_{t' \neq t, t'=1}^{T} \sum_{t=1}^{T} \sqrt{\mathrm{Var}\left[ \nabla \frac{w_t^1}{Z_t} \right] \mathrm{Var}\left[ \nabla \frac{w_{t'}^1}{Z_{t'}} \right]} \right\} + \mathcal{O}(T^2/K^2)}} \right| \tag{35}
$$

*where $Z = p_\theta(\mathbf{x}_{1:T})$ and $Z_t = p_\theta(\mathbf{x}_t|\mathbf{x}_{1:t-1})$ for $t \in 1 : T$.*

*Further assuming the resampling bias $\sum_{t=2}^{T} \sum_{t' \geq t+1}^{T} \mathbb{E}\left[ \nabla \frac{w_{t-1}^1}{Z_{t-1}} \cdot \frac{(w_{t'}^1 - Z_{t'})^2}{2Z_{t'}^2} \middle| (a_{t-1}^1 = 1) \right] = \mathcal{O}(1)$ leads to $SNR_K(\theta, \phi, \varphi) = \mathcal{O}(\sqrt{K})$.*

The formal proof is provided in the Appendix. We add empirical evidence to this result in Section 4. For the rest of experiments, we use $\nabla \mathcal{L}_K$ to estimate the gradient.

## 4 EXPERIMENTAL RESULTS

It is important to quantify the performance of the hidden evolution in order to understand the quality of the dynamics learned. The $k$-step MSE and its normalized version, the $R^2_k$ are computed by applying the transition function to the system over a rolling window $k$ steps into the future without any input data. The emission function is then used to form a predicted reconstruction $\hat{\mathbf{x}}_{t+k}$ and an error term using the corresponding observation window $\mathbf{x}_{t+k}$ (below $\bar{\mathbf{x}}_k$ is the average of $\mathbf{x}_{k:T}$). This ensures that the evaluation criteria is dependent upon the both evolution term as well as the emission term.

$$
\mathrm{MSE}_k = \sum_{t=0}^{T-k} (\mathbf{x}_{t+k} - \hat{\mathbf{x}}_{t+k})^2 \qquad R^2_k = 1 - \frac{\mathrm{MSE}_k}{\sum_{t=0}^{T-k} (\mathbf{x}_{t+k} - \bar{\mathbf{x}}_k)^2} \tag{36}
$$

We note that the ELBO is not a performance statistic that generalizes across models. In contrast, the $R^2_k$ provides a metric to quantify the inferred dynamics. Fig. 6 in the Appendix emphasizes the above by highlighting the difference between the ELBO and the $R^2_k$ for the IWAE and SVO.

### 4.1 FITZHUGH-NAGUMO

The Fitzhugh-Nagumo system is a two dimensional reduction of the Hodgkin-Huxley model. It is described by two independent variables with cubic and linear functions.

$$
\dot{V} = V - V^3/3 - W + I_{ext},
$$
$$
\dot{W} = a(bV - cW) \tag{37}
$$

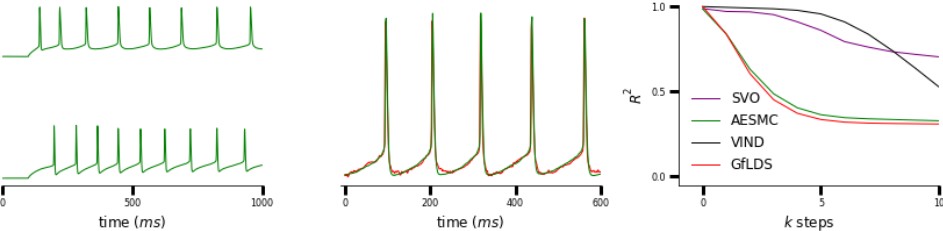

Figure 4: Summary of the Allen results: (left) two trials from the dataset; (center) the data against the predicted observation value using the dynamics learned over a rolling window ten steps ahead. Hyperpolarization and depolarization nonlinearities are predicted by the inferred dynamics; (right) $R_k^2$ for various models.

The system was integrated over 200 time points with $I_{ext} = 1$ held constant and $a = 0.7, b = 0.8, c = 0.08$. The initial state was sampled uniformly over $[-3, 3]^2$ and 100 trials were generated with 66 for training, 17 for validation and 17 for testing. One dimensional Gaussian observations were produced with a linear observation model acting on a single variable. Dimensionality expansion is intrinsically harder than dimensionality reduction due to the loss of information. For this reason we choose $\mathbf{z}_t = (V_t, W_t)$ and $\mathbf{x}_t = \mathcal{N}(V_t, \sigma^2)$ where $\sigma^2 = 0.01$.

Fig. 1 shows the results of the Fitzhugh-Nagumo experiment. The top left panel displays the phase space and trajectories of the original system. The bottom left panel displays latent variables from a single trial in the plane. The top center panel displays the inferred two-dimensional dynamics and trajectories. We find that the limit cycle is recovered correctly and the topology of the space is preserved. Initial points located both inside and outside of the cycle in the original system are invariant in the reconstruction. The bottom center panel displays an observation from a single trial. The right panel shows the $R_k^2$ comparison across models.

## 4.2    Elbo Convergence with Shared Evolution

We explore the effect of sharing the evolution parameters between the proposal and target distribution on model convergence by examining the ELBO. Fig. 2 displays the ELBO convergence across epochs as we increase the number of particles ($K$). The left panel illustrates the ELBO convergence sharing network parameters $\varphi = \{\psi, \Sigma\}$ between proposal and target. Larger values of $K$ result in a faster convergence and lower stochastic gradient noise. The center panel displays the ELBO convergence for separate evolution networks for the proposal and the target. Note that for the same value of $K$, separate evolution networks require a larger number of epochs to converge and the ELBO obtains a lower value with larger stochastic gradient noise. The right panel displays both shared and independent evolution terms for $K = 16$ particles highlighting the difference between the two parameterizations.

## 4.3    Lorenz Attractor

The Lorenz attractor is a chaotic nonlinear dynamical system defined with 3 independent variables.

$$\dot{z}_1 = \sigma(z_2 - z_1), \qquad \dot{z}_2 = z_1(\rho - z_3) - z_2, \qquad \dot{z}_3 = z_1 z_2 - \beta z_3 \qquad (38)$$

The system of equations is integrated over 250 time points using $\sigma = 10, \rho = 28, \beta = 8/3$ by generating randomized initial states in $[-10, 10]^3$. Ten dimensional Gaussian observations were produced with a mean specified by a $\mathbf{z}$-dependent neural network. The final dataset consists of 100 trials with 66 for training, 17 for validation and 17 for testing.

The results of the Lorenz experiment are shown in Fig. 3. The inferred latent paths are put together in the left panel illustrating the two cycles. The center panel displays how $R_{k=10}^2$ scales as the latent dimension $d_{\mathbf{z}}$ increases. The $R_k^2$ stops improving once the correct number of independent variables in the system is reached. The right panel displays the $R_k^2$ comparison with $d_{\mathbf{z}} = 3$.

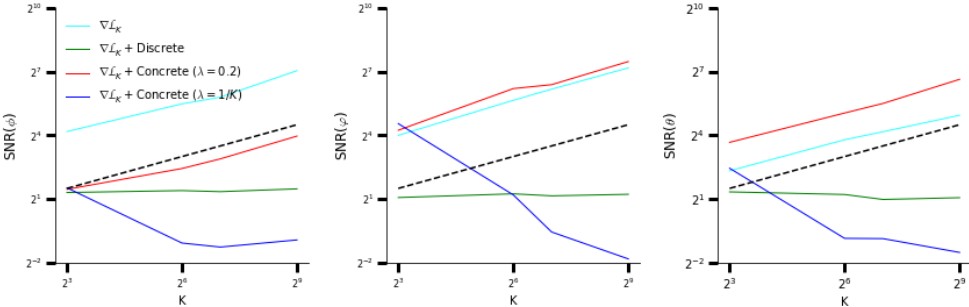

Figure 5: Convergence of SNRs of gradient estimators in the encoder network (left), transition network (center) and decoder network (right) with increasing $K$. Distinct lines correspond to gradient estimators. The black dashed line with slope 1 illustrates a signal-to-noise-ratio of convergence rate $\mathcal{O}(\sqrt{K})$.

### 4.4 SINGLE CELL ELECTROPHYSIOLOGY DATA

Electrophysiology data was downloaded from the Allen Brain Atlas (Jones et al., 2009). Intracellular voltage recordings were collected from primary Visual Cortex of the mouse, area layer 4. The dataset consists of 40 trials from 5 different cells. The input current in the experiments is a step-function with an amplitude between 80 and 151pA. There were 30 trials reserved for training and 10 for validation. Each trial was divided into five parts and down-sampled from 10,000 time bins to 1,000 in equal-time intervals. Trials were normalized dividing each by its maximal value.

Fig. 4 displays the result of the Allen experiment. The left panel illustrates two trials of 1D observations from the training set. We find that three latent variables provides the best fit. The center panel displays the predicted observation value using the dynamics learned over a rolling window ten steps ahead. Hyperpolarization and depolarization nonlinearities are accurately reconstructed by appying the inferred dynamics. The right panel displays the $R_k^2$ comparison with $d_{\mathbf{z}} = 3$.

### 4.5 SIGNAL-TO-NOISE RATIO OF GRADIENT ESTIMATORS

In this section, we focus on SNRs of gradient estimators in the encoder network ($\phi$), evolution network ($\varphi$) and decoder network ($\theta$), where the gradient is taken with respect to $\phi$, $\varphi$ and $\theta$ correspondingly. Fig. 5 presents the empirical results for SNRs of four different gradient estimators. The expectation and variance in the SNR are calculated using $N = 100$ gradient samples, which are collected in the middle training stage of running SVO on Fitzhugh-Nagumo data. The $l_2$ norm is used to compute a scalar quantity to define the SNR. We find that the three panels share similar patterns: (1) The gradient estimator without resampling and the relaxed one with temperature $\lambda = 0.2$ possess an SNR of convergence rate $\mathcal{O}(\sqrt{K})$, which aligns with the theoretical result. Although ignoring or relaxing the gradient from the resampling procedure adds a bias to the gradient estimator, it produces a positive effect of $K$ on SNR; (2) The unbiased CATEGORIAL resampling gradient and relaxed gradient with small bias ($\lambda = (K-1)^{-1}$) suffer from large variance which increases at a rate of $\mathcal{O}(\sqrt{K})$, leading to a relatively low and even decreasing SNR for increasing $K$.

## 5 CONCLUSION

We have sketched a method to construct Smoothed Variational Objectives (SVOs) using Sequential Monte Carlo to perform both inference and parameter estimation in nonlinear dynamical systems. By sharing parameters of the transition function between the proposal and target, SVO forgoes exchanging expressiveness for tractability. We find that the use of a biased gradient estimate can help address the issue of a degraded signal-to-noise ratio. Using the smoothed proposal density improves the computational complexity with a cost independent of the number of particles, and the prediction performance, quantified by the k-step reconstruction error across dimensionality reduction and expansion tasks. Future work includes parameterizing an input term to model non-autonomous dynamical systems.

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

APPENDIX

## A. IWAE vs SVO COMPARISON

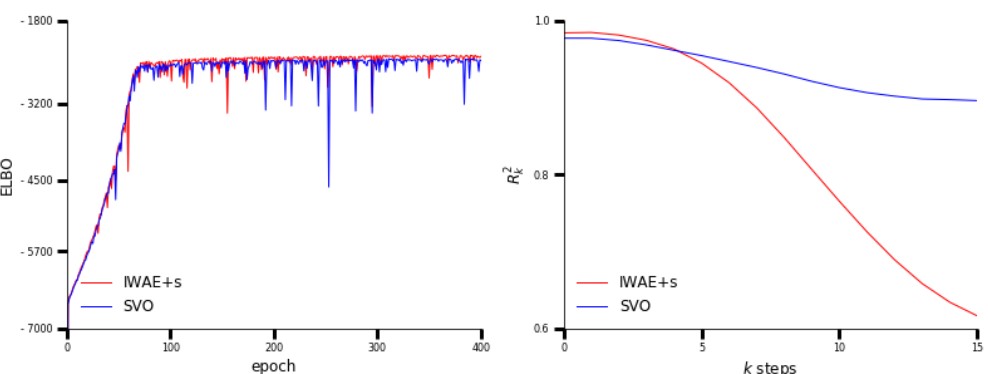

Figure 6: Summary of the IWAE vs SVO comparison on the Lorenz task: (left) ELBO across epochs for the smoothed IWAE vs SVO. The latter exhibits larger stochastic gradient noise due to the resampling process; (right) $R_k^2$ for the smoothed IWAE vs SVO on the test set.

The smoothed IWAE achieves a higher ELBO and lower zero-step reconstruction error than SVO, however the $R_k^2$ is significantly lower illustrating the inability of the dynamics learned to predict future observations. We emphasize the relevance of the $R_k^2$ as opposed to the ELBO in quantifying the inferred dynamics. In the comparison above, both models were trained using three layer networks for encoder, shared transition and decoder functions and using $64$ particles.

## B. GRADIENT ESTIMATORS

1. The unbiased gradient estimator,
   $\nabla \mathcal{L}_K + \text{CATEGORICAL} = \nabla \log \prod_{t=1}^{T-1} \prod_{k=1}^{K} \text{CATEGORICAL}(a_t^{(k)}|w_t^{1:K}) \cdot \mathcal{L}_K + \nabla \mathcal{L}_K$.
   The derivation could be found in (Le et al., 2018) or (Maddison et al., 2017).

2. The biased gradient estimator without resampling $\nabla \mathcal{L}_K$ is implemented by simply taking the gradient of the estimated variational objective, $\hat{Z}_{\text{SVO}}$.

3. For the relaxed gradient estimator,
   $\nabla \mathcal{L}_K + \text{CONCRETE}(\lambda) = \nabla \log \prod_{t=1}^{T-1} \prod_{k=1}^{K} \text{CONCRETE}(a_t^{(k)}|w_t^{1:K}, \lambda) \cdot \mathcal{L}_K + \nabla \mathcal{L}_K$
   We use the CONCRETE distribution to resample particles, and then directly evaluate the gradient of the objective.

## C. PROOF OF PROPOSITION 1

*Proof.* It suffices to show the convergence rate of expectation and variance of gradient estimate with respect to $K$. Throughout the analysis, we will extensively apply the result from (Rainforth et al., 2018), and exploit the factorization of the SVO objective: $\hat{Z} = \prod_{t=1}^{T} \hat{Z}_t$ where $\hat{Z}_t = \frac{1}{K} \sum_{k=1}^{K} w_t^k$. Assume that $\mathbf{z}_{1:T}^{1:K}$ are obtained by passing the Guassian noise $\epsilon_{1:T}^{1:K}$ through the reparameterization function.

1. Expectation.

$$\mathbb{E}\left[\nabla \log \hat{Z}\right] = \nabla \mathbb{E}\left[\log \hat{Z}\right] - \mathbb{E}\left[\nabla \log \prod_{t=2}^{T} \prod_{k=1}^{K} \text{CATEGORICAL}(a_{t-1}^k|w_{t-1}^{1:K}) \cdot \log \hat{Z}\right]$$

(39)

The expectation decomposes into two terms, where the convergence rate for the first directly follows the result from (Rainforth et al., 2018):

$$\nabla \mathbb{E}\left[\log \hat{Z}\right] = \nabla \sum_{t=1}^{T} \mathbb{E}\left[\log \hat{Z}_t\right] \tag{40}$$

$$= \nabla \log Z - \frac{1}{2K}\left[\sum_{t=1}^{T} \nabla(\mathrm{Var}[w_t^1]/z_t^2)\right] + \mathcal{O}(T/K^2) \tag{41}$$

For the remaining term that includes the resampling gradient, we apply a thorough analysis as follows.

$$\mathbb{E}\left[\nabla \log \prod_{t=2}^{T}\prod_{k=1}^{K} \textsc{Categorical}(a_{t-1}^k|w_{t-1}^{1:K}) \cdot \log \hat{Z}\right]$$

$$= \sum_{t=2}^{T}\sum_{k=1}^{K} \mathbb{E}\left[\nabla \log \textsc{Categorical}(a_{t-1}^k|w_{t-1}^{1:K}) \cdot \log \hat{Z}\right] \tag{42}$$

$$= K\sum_{t=2}^{T}\sum_{t'=1}^{T} \mathbb{E}\left[\nabla \log \textsc{Categorical}(a_{t-1}^1|w_{t-1}^{1:K}) \cdot \log \hat{Z}_{t'}\right] \tag{43}$$

Taylor expand $\log \hat{Z}_{t'}$ about $Z_{t'}$:

$$= K\sum_{t=2}^{T}\sum_{t'=2}^{T} \mathbb{E}\Big[\nabla \log \textsc{Categorical}(a_{t-1}^1|w_{t-1}^{1:K})$$

$$\cdot \Big(\log Z_{t'} + \frac{\hat{Z}_{t'} - Z_{t'}}{Z_{t'}} - \frac{(\hat{Z}_{t'} - Z_{t'})^2}{2Z_{t'}^2} + R_3(\hat{Z}_{t'})\Big)\Big] \tag{44}$$

where $R_3(\hat{Z}_{t'})$ denotes the remainder in the Taylor expansion of $\log \hat{Z}_{t'}$ about $Z_{t'}$.

For $t' \leq t - 1$, we have

$$\mathbb{E}\left[\nabla \log \textsc{Categorical}(a_{t-1}^1|w_{t-1}^{1:K}) \cdot \frac{(\hat{Z}_{t'} - Z_{t'})}{Z_{t'}}\right]$$

$$= \mathbb{E}_{\epsilon_{1:t-1}^{1:K}, a_{1:t-2}^{1:K}}\left[\frac{\hat{Z}_{t'} - Z_{t'}}{Z_{t'}} \cdot \mathbb{E}_{a_{t-1}^1}\left[\nabla \log \textsc{Categorical}(a_{t-1}^1|w_{t-1}^{1:K})\right]\right] \tag{45}$$

$$= \mathbb{E}_{\epsilon_{1:t-1}^{1:K}, a_{1:t-2}^{1:K}}\left[\frac{\hat{Z}_{t'} - Z_{t'}}{Z_{t'}} \cdot 0\right] = 0.$$

For $t' \geq t$, we have

$$\mathbb{E}\left[\nabla \log \textsc{Categorical}(a_{t-1}^1|w_{t-1}^{1:K}) \cdot \frac{(\hat{Z}_{t'} - Z_{t'})}{Z_{t'}}\right]$$

$$= \mathbb{E}_{\epsilon_{1:t-1}^{1:K}, a_{1:t-1}^{1:K}}\left[\nabla \log \textsc{Categorical}(a_{t-1}^1|w_{t-1}^{1:K}) \cdot \mathbb{E}_{\epsilon_{t:t'}^{1:K}, a_{t:t'-1}^{1:K}}\left[\frac{\hat{Z}_{t'} - Z_{t'}}{Z_{t'}}\right]\right]$$

$$= \mathbb{E}_{\epsilon_{1:t-1}^{1:K}, a_{1:t-1}^{1:K}}\left[\nabla \log \textsc{Categorical}(a_{t-1}^1|w_{t-1}^{1:K}) \cdot 0\right] = 0 \tag{46}$$

Hence, it suffices to compute the convergence rate of the following:

$$K\sum_{t=2}^{T}\sum_{t'=2}^{T} \mathbb{E}\left[\nabla \log \textsc{Categorical}(a_{t-1}^1|w_{t-1}^{1:K}) \cdot \frac{(\hat{Z}_{t'} - Z_{t'})^2}{2Z_{t'}^2}\right] \tag{47}$$

Note that when $t' \leq t - 1$, we obtain similar results as Eq. (45). Thus, we turn to the case when $t' \geq t$.

For $t' \geq t + 1$, each $w_{t'}^k$ has dependence on $a_{t-1}^1$, hence

$$K \cdot \mathbb{E}\left[\nabla \log \text{CATEGORIAL}(a_{t-1}^1 | w_{t-1}^{1:K}) \cdot \frac{(\hat{Z}_{t'} - Z_{t'})^2}{2Z_{t'}^2}\right] \tag{48}$$

$$= K \cdot \mathbb{E}\left[\nabla \log \text{CATEGORICAL}(a_{t-1}^1 | w_{t-1}^{1:K}) \cdot \frac{\left(1/K \sum_{k=1}^K (w_{t'}^k - Z_{t'})\right)^2}{2Z_{t'}^2}\right] \tag{49}$$

$$= \mathbb{E}\left[\nabla \log \text{CATEGORICAL}(a_{t-1}^1 | w_{t-1}^{1:K}) \cdot \frac{(w_{t'}^1 - Z_{t'})^2}{2Z_{t'}^2}\right] \tag{50}$$

$$= \sum_{i=1}^K \mathbb{E}_{\epsilon_{1:t-1}^{1:K} a_{1:t-2}^{1:K}}\left[\mathbb{E}_{\epsilon_t^1}\left[\nabla \frac{w_{t-1}^1}{K\hat{Z}_{t-1}} \cdot \frac{(w_t^1 - Z_t)^2}{2Z_t^2}\middle| (a_{t-1}^1 = i)\right]\right] \tag{51}$$

$$= K \cdot \mathbb{E}_{\epsilon_{1:t-1}^{1:K} a_{1:t-2}^{1:K}}\left[\mathbb{E}_{\epsilon_t^1}\left[\nabla \frac{w_{t-1}^1}{K\hat{Z}_{t-1}} \cdot \frac{(w_t^1 - Z_t)^2}{2Z_t^2}\middle| (a_{t-1}^1 = 1)\right]\right] \tag{52}$$

Applying the Taylor expansion of $\frac{1}{\hat{Z}_{t-1}}$ around $Z_{t-1}$: $\frac{1}{\hat{Z}_{t-1}} = \frac{1}{Z_{t-1}} + R_2(\hat{Z}_{t-1})$

$$= \mathbb{E}_{\epsilon_{1:t-1}^{1:K} a_{1:t-2}^{1:K}}\left[\mathbb{E}_{\epsilon_t^1}\left[\nabla \frac{w_{t-1}^1}{Z_{t-1}} \cdot \frac{(w_{t'}^1 - Z_{t'})^2}{2Z_{t'}^2}\middle| (a_{t-1}^1 = 1)\right]\right]$$
$$+ \mathbb{E}_{\epsilon_{1:t-1}^{1:K} a_{1:t-2}^{1:K}}\left[\mathbb{E}_{\epsilon_t^1}\left[\nabla(w_{t-1}^1 R_2(\hat{Z}_{t-1})) \cdot \frac{(w_{t'}^1 - Z_{t'})^2}{2Z_{t'}^2}\middle| (a_{t-1}^1 = 1)\right]\right] \tag{53}$$

For $t' = t$, only $w_t^1$ depends on $a_{t-1}^1$, consequently we have

$$K \cdot \mathbb{E}\left[\nabla \log \text{CATEGORICAL}(a_{t-1}^1 | w_{t-1}^{1:K}) \cdot \frac{(\hat{Z}_{t'} - Z_{t'})^2}{2Z_{t'}^2}\right] \tag{54}$$

$$= \frac{1}{K} \cdot \mathbb{E}_{\epsilon_{1:t-1}^{1:K} a_{1:t-2}^{1:K}}\left[\mathbb{E}_{\epsilon_t^1}\left[\nabla \frac{w_{t-1}^1}{Z_{t-1}} \cdot \frac{(w_{t'}^1 - Z_{t'})^2}{2Z_{t'}^2}\middle| (a_{t-1}^1 = 1)\right]\right]$$
$$+ \frac{1}{K} \cdot \mathbb{E}_{\epsilon_{1:t-1}^{1:K} a_{1:t-2}^{1:K}}\left[\mathbb{E}_{\epsilon_t^1}\left[\nabla(w_{t-1}^1 R_2(\hat{Z}_{t-1})) \cdot \frac{(w_{t'}^1 - Z_{t'})^2}{2Z_{t'}^2}\middle| (a_{t-1}^1 = 1)\right]\right] \tag{55}$$

2. Variance.

$$\text{Var}\left[\nabla \log \hat{Z}\right] = \text{Var}\left[\sum_{t=1}^T \nabla \log \hat{Z}_t\right] \tag{56}$$

$$= \sum_{t=1}^T \text{Var}\left[\nabla \log \hat{Z}_t\right] + 2\sum_{t=1}^T \sum_{t' \neq t, t'=1}^T \text{Cov}\left(\nabla \log \hat{Z}_t, \nabla \log \hat{Z}_{t'}\right) \tag{57}$$

Decomposing the variance into the sum of variance at each time points, and the pairwise covariance across different time point, we will show that both terms are $\mathcal{O}(1/K)$.

(1) Variance at each time step. $\forall t = 1 : T$,

$$\text{Var}\left[\nabla \log \hat{Z}_t\right] = \frac{1}{K} \cdot \mathbb{E}\left[\left(\frac{Z_t \nabla w_t^1 - w_t^1 \nabla Z_t}{Z_t^2}\right)^2\right] + \mathcal{O}(1/K^2) \tag{58}$$

$$= \frac{1}{K} \cdot \mathbb{E}\left[\left(\frac{\nabla w_t^1}{Z_t}\right)^2\right] + \mathcal{O}\left(1/K^2\right) \tag{59}$$

(2) Covariance between different time steps.

For $t \neq t' \in 1 : T$, we first apply Taylor theorem to $\log \hat{Z}_t$ around $Z_t$, and then exploit the fact that $\hat{Z}_t$ is an unbiased estimation of $Z_t$, and exploit the definition of covariance to expand and collapse terms, as follows:

$$\text{Cov}\left(\nabla \log \hat{Z}_t, \nabla \log \hat{Z}_{t'}\right) \tag{60}$$

$$= \text{Cov}\left(\nabla\left(\log Z_t + \frac{\hat{Z}_t - Z_t}{Z_t} + R_1(\hat{Z}_t)\right), \nabla\left(\log Z_{t'} + \frac{\hat{Z}_{t'} - Z_{t'}}{Z_{t'}} + R_1(\hat{Z}_{t'})\right)\right)$$

$$= \text{Cov}\left(\nabla\left(\frac{\hat{Z}_t - Z_t}{Z_t} + R_1(\hat{Z}_t)\right), \nabla\left(\frac{\hat{Z}_{t'} - Z_{t'}}{Z_{t'}} + R_1(\hat{Z}_{t'})\right)\right) \tag{61}$$

$$= \mathbb{E}\left[\nabla\left(\frac{\hat{Z}_t}{Z_t}\right) \cdot \nabla\left(\frac{\hat{Z}_{t'}}{Z_{t'}}\right)\right] + \mathbb{E}\left[\nabla\left(\frac{\hat{Z}_{t'}}{Z_{t'}}\right) \cdot \nabla R_1(Z_t)\right]$$

$$+ \mathbb{E}\left[\nabla\left(\frac{\hat{Z}_t}{Z_t}\right) \cdot \nabla R_1(Z_{t'})\right] + \text{Cov}\left(\nabla R_1(\hat{Z}_t), \nabla R_1(\hat{Z}_{t'})\right) \tag{62}$$

(i) For the first term in Eq. (62), since $\mathbf{z}_t^{(k)}$ are i.i.d. for fixed $t$, we have

$$\mathbb{E}\left[\nabla\left(\frac{\hat{Z}_t}{Z_t}\right) \cdot \nabla\left(\frac{\hat{Z}_{t'}}{Z_{t'}}\right)\right] = \mathbb{E}\left[\frac{1}{K}\sum_{k=1}^{K}\nabla\left(\frac{w_t^k}{Z_t}\right) \cdot \frac{1}{K}\sum_{k'=1}^{K}\nabla\left(\frac{w_{t'}^{k'}}{Z_{t'}}\right)\right] \tag{63}$$

$$= \frac{1}{K^2} \cdot \sum_{k=1}^{K}\sum_{k'=1}^{K}\mathbb{E}\left[\nabla\left(\frac{w_t^k}{Z_t}\right) \cdot \nabla\left(\frac{w_{t'}^{k'}}{Z_{t'}}\right)\right] \tag{64}$$

$$= \mathbb{E}\left[\nabla\frac{w_t^1}{Z_t} \cdot \nabla\frac{w_{t'}^1}{Z_{t'}}\right] \tag{65}$$

$$= \text{Cov}\left(\nabla\frac{w_t^1}{Z_t}, \nabla\frac{w_{t'}^1}{Z_{t'}}\right) \tag{66}$$

Without loss of generality, we assume $t' > t$. First, when $t' = t + 1$,

$$\Pr\left(\mathbf{z}_{t+1}^1 \text{ depends on } \mathbf{z}_t^1\right) = \mathbb{E}\left[\frac{w_t^1}{\sum_{k=1}^{K} w_t^k}\right] = \frac{1}{K} \tag{67}$$

When $t' > t + 1$, using chain rule and by induction we also have,

$$\Pr(\mathbf{z}_{t'}^1 \text{ depends on } \mathbf{z}_t^1) = \frac{1}{K} \tag{68}$$

Hence,

$$\text{Cov}\left(\nabla\frac{w_t^1}{Z_t}, \nabla\frac{w_{t'}^1}{Z_{t'}}\right) = \frac{1}{K} \cdot \text{Cov}\left(\nabla\frac{w_t^1}{Z_t}, \nabla\frac{w_{t'}^1}{Z_{t'}}\middle| \left(z_{t'}^1 \text{ depends on } z_t^1\right)\right) \tag{69}$$

$$\leq \frac{1}{K}\sqrt{\text{Var}\left[\nabla\frac{w_t^1}{Z_t}\right]\text{Var}\left[\nabla\frac{w_{t'}^1}{Z_{t'}}\right]} \tag{70}$$

(ii) For the second and third term in Eq. (62), without loss of generality, we analyze the second term $\mathbb{E}\left[\nabla\left(\hat{Z}_{t'}/Z_{t'}\right) \cdot \nabla R_1(Z_t)\right]$, and assume $t' > t$.
Using the i.i.d. property of particles at fixed time step, we have

$$\mathbb{E}\left[\nabla\left(\frac{\hat{Z}_{t'}}{Z_{t'}}\right) \cdot \nabla R_1(Z_t)\right] = \frac{1}{K^3} \cdot \mathbb{E}\left[\sum_{k=1}^{K}\nabla\frac{w_{t'}^k}{Z_{t'}}\mathcal{O}\left(\sum_{k=1}^{K}(w_t^k - Z_t)^2\right)\right] \tag{71}$$

$$= \frac{1}{K} \cdot \mathbb{E}\left[\nabla\frac{w_{t'}^1}{Z_{t'}}\mathcal{O}\left((w_t^1 - Z_t)^2\right)\right] \tag{72}$$

Similar to the previous analysis on covariance, we can show that

$$\mathbb{E}\left[\nabla \frac{w_{t'}^1}{Z_{t'}} \cdot \mathcal{O}\left((w_t^1 - Z_t)^2\right)\right] = \mathcal{O}\left(\frac{1}{K}\right) \tag{73}$$

Hence,

$$\mathbb{E}\left[\nabla\left(\frac{\hat{Z}_{t'} - Z_{t'}}{Z_{t'}}\right) \cdot \nabla R_1(Z_t)\right] = \mathcal{O}\left(\frac{1}{K^2}\right) \tag{74}$$

(iii) For the last term in Eq. (62), note that $|\mathrm{Cov}(A, B)| \leq \sqrt{\mathrm{Var}(A)\mathrm{Var}(B)}$, and $\mathrm{Var}[\nabla R_1(\hat{Z}_t)] = \mathcal{O}\left(1/K^2\right)$, hence we obtain

$$\mathrm{Cov}\left(\nabla R_1(\hat{Z}_t), \nabla R_1(\hat{Z}_{t'})\right) = \mathcal{O}\left(\frac{1}{K^2}\right) \tag{75}$$

Substituting Eq. (59), Eq. (62) and Eq. (66) into Eq. (57), we arrive at the final expression for the variance of gradient estimate:

$$\mathrm{Var}\left[\nabla \log \hat{Z}\right] \tag{76}$$

$$= \frac{1}{K}\left\{\sum_{t=1}^{T}\mathbb{E}\left[\left(\nabla \frac{w_t^1}{Z_t}\right)^2\right] + \sum_{t'\neq t,t'=1}^{T}\sum_{t=1}^{T}\sqrt{\mathrm{Var}\left[\nabla \frac{w_t^1}{Z_t}\right]\mathrm{Var}\left[\nabla \frac{w_{t'}^1}{Z_{t'}}\right]}\right\} + \mathcal{O}\left(\frac{T^2}{K^2}\right)$$

$\square$

