# OpenReview forum: "Smoothing Nonlinear Variational Objectives with Sequential Monte Carlo"
_ICLR.cc/2019/Workshop/DeepGenStruct — DeepGenStruct 2019_

### Official Review · AnonReviewer1 · 2019-04-13
**Not new, not smoothing, still an interesting question**

**Rating:** 2
**Confidence:** 3

**Review:**

This paper proposes two methods to extend recent work in filtering SMC-based variational objectives to the smoothing case. The first approach is a Monte Carlo objective (MCO) based on Forward Filtering Backward Smoothing (FFBS), and the second technique gives the SMC proposal distribution access to all observations. The authors evaluate both techniques experimentally and find that the FFBS-based technique does not perform well, while giving the proposal access to all observations improves performance on several tasks.

While the paper is generally written well and easy to follow, the main technique (giving the proposal access to all observations) has been presented previously in the literature and is misrepresented as a smoothing algorithm when it is not.

To expand on these points:

1. Filtering Variational Objectives (Maddison et al. 2017) section 6.4 presents experiments that run FIVO with a proposal that conditions on the state of a bidirectional RNN run over the observations. They find that it does not reliably help on their tasks.
2. Changing the information that the proposal distribution has access to does not change SMC’s sequence of target distributions. If only the proposal is changed and the form of the weights is not changed, then the algorithm is still based on filtering SMC and is not smoothing. Unfortunately, it is not entirely clear what SMC scheme the authors use with the new proposal. It seems that they use the future-conditioned proposal with the weights defined in equation (12), but if this is not the case it should be clarified.

Further feedback:

1. As discussed in Maddison et al. 2017, an MCO based on filtering SMC cannot become tight, even when q is set to the true smoothing distribution. Because of this, it is useful to compare the performance of the proposed algorithm to the IWAE bound (with the same proposal) which can become tight and allow the proposal to make full use of the information available to it. The authors should consider incorporating this comparison in their experiments.
2. On page 4 the authors state “As with the IWAE, increasing K yields a tighter bound L_SMC defined below”. I am not aware of a proof that L_SMC is provably tighter as K increases. The authors should provide a proof or citation or remove the statement.
3. There is prior work on developing variational objectives based on smoothing SMC, including

Graphical model inference: Sequential Monte Carlo meets deterministic approximations, Lindsten et al 2018
Twisted Variational Sequential Monte Carlo, Lawson et al. 2018

The authors should consider incorporating this in their related works.

Overall, it is still interesting to consider why an MCO based on filtering SMC would perform better when the proposal is given access to all observations. If the authors change their paper to address the points above, I will consider changing my score.

---

### Official Review · AnonReviewer2 · 2019-04-17

**Rating:** 4
**Confidence:** 1

**Review:**

This paper describes a framework called Smoothed Variational Objectives (SVOs) for performing inference and parameter estimation in nonlinear dynamical systems. The proposed method is evaluated on three benchmarks (Fitzhugh-Nagumo, Lorenz Attractor, and electrophysiology data) in terms of R^2_k, and shows favorable results compared to previous algorithms.

Overall, I think this paper is well-written and well-structured. It provides enough background on variational inference and Sequential Monte Carlo methods and is more or less self-contained. Unfortunately, I am not an expert on this topic and won’t be able to provide more insightful opinions.

Other questions/comments:
Is the \delta term in Eq defined anywhere?

---

### Decision · Program_Chairs · 2019-04-19
**Acceptance Decision**

**Decision:**

Accept

**Comment:**

This paper  studies sequential MC for training deep generative models. The paper is well-written but the authors should connect the work to existing works as mentioned by reviewer 1